# Using Enhanced Gaussian Cross-Entropy in Imitation Learning to Digging the First Diamond in Minecraft

**Yingjie Cai & Xiao Zhang**
Department of Electronic Engineering
The Chinese University of Hong Kong
Shatin, Hong Kong
`{caiyingjie,xzhang9411}@link.cuhk.edu.hk`

## Abstract

Although state-ofthe-art reinforcement learning (RL) systems has led to break-throughs in many difficult tasks, the sample inefficiency of standard reinforcement learning methods still precludes their application to more extremely complex tasks. Such limitation will make many reinforcement learning systems cannot be applied to real-world problem, in which environment samples are expensive. To solve this problem, MineRL (13) provide an ideal develop environment to facilitate the research that leveraging fewer human demonstrations with more efficient reinforcement learning systems. Based on the MineRL environmnet, we proposed an enhanced Gaussian cross entropy (EGCE) loss for imitation learnning problems to achieve ideal performance. In the **ObtainDiamond** task, our EGCE achieves about 7.7% improvement than a strong baseline imitation learning pipeline. The demo video is available at *here*.

## 1 Introduction

As deep reinforcement learning (DRL) methods are applied to increasingly difficult as well as complex problems, the number of samples used for training increases: AlphaGo uses about 5 million games of self-play (27), while AlphaStar uses over 200 years of StarCraft II (3). Moreover, OpenAI Five spend more than $11,000$ years of Dota2 gameplay (4).

### 1.1 Task Formulation

Although there exist many data augmentation methods and designed real-world environments for limited numbers of trials, these methods are remain not sufficiently sample efficient for a large part of complex real-world domains. Therefore an effective environment is necessary for such situation. In MineRL (13), a large scale dataset that contains over 60 million state-action pairs of human demonstrations, several related tasks is designed with Minecraft game environment.

The main task in MineRL (13) solving the **ObtainDiamond** environment (12; 14) in MineCraft. Minecraft is a 3D, first-person, open-world game that centered around the resource gathering and items/structures creation. These structures and items have prerequisite tools and materials required for their creation. As a result, many items require the completion of a series of natural subtasks. Solving the **ObtainDiamond** environment consists of controlling an embodied agent to obtain a diamond by navigating the complex item hierarchy of Minecraft. In solving this task, a learning algorithm has direct access to a $64 \times 64$ pixel observation from the perspective of the embodied Minecraft agent, and a set of discrete observations consisting of each item required for obtaining a diamond that the agent has in its possession. The action space is the Cartesian product of continuous view adjustment (turning and pitching), binary movement commands (left/right, forward/backward), and discrete actions for placing blocks, crafting items, smelting items, and mining/hitting enemies. An

The stages of obtaining a diamond.

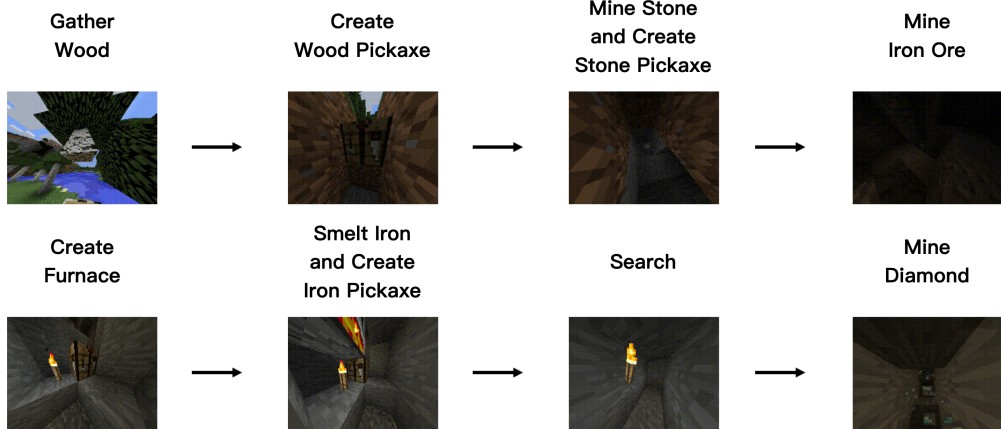

Figure 1: These stages exhibit the different periods that a agent/player gets the first diamond in MineCraft **ObtainDiamond** environment.

agent receives reward for completing the full task of obtaining a diamond. The full task of obtaining a diamond can be decomposed into a sequence of prerequisite subtasks of increasing difficulty. An agent also receives reward for the first time it accomplishes each subtask in the sequence. An agent receives twice the reward as received for accomplishing the previous subtask (starting from a reward of 1). The exception to this rule is achieving the full ObtainDiamond task by obtaining a diamond: accomplishing the final task is worth four times as much as completing the previous subtask.

## 1.2 Environments and Dataset

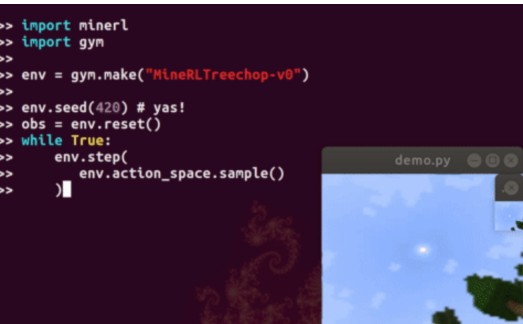

Figure 2: Setting an environment in MineRL.

MineRL (13) define one primary **ObtainDiamond** environment, and six other auxiliary environments (Navigate, Treechop, ObtainCookedMeat, ObtainBed, and ObtainIronPickaxe) that encompass a significant portion of human Minecraft play. By training agent in these environment domains, many of the hardest challenges in reinforcement learning, such as sparse rewards, long reward horizons, and efficient hierarchical planning can be completely revealed. The main task is solving the **ObtainDiamond** environment, where the agent begins in a random initializing location without any items or structures, and is tasked with obtaining one diamond. The agent receives a high reward for obtaining a diamond as well as smaller rewards for obtaining prerequisite items. Episodes will end once the agent dying, successfully obtaining a diamond, or reaching the maximum step limitation of $18,000$ frames ( about 800 seconds). The **ObtainDiamond** environment is a difficult environment. Diamonds only exist in a small portion of the world and are 2-10 times rarer than other ores in

| Milestone | Reward | Milestone | Reward |
|-----------|--------|-----------|--------|
| log | 1 | furnace | 32 |
| planks | 2 | stone_pickaxe | 32 |
| stick | 4 | iron_ore | 64 |
| crafting_table | 4 | iron_ingot | 128 |
| wooden_pickaxe | 8 | iron_pickaxe | 256 |
| stone | 16 | diamond | 1024 |

Table 1: Rewards for achieving sub-goals and main goal (diamond) for **ObtainDiamond**.

| Baseline Method | Reward | Type |
|-----------------|--------|------|
| SQIL (22) | 2.94 | Imitation Learning |
| Rainbow (15) | 0.35 | Reinforcement Learning |
| DQFD (29) | 0.24 | Imitation Learning with Reinforcement Learning |
| PDDDQN (26) | 0.11 | Reinforcement Learning |

Table 2: Some baselines of **ObtainDiamond** environment.

Minecraft. Additionally, obtaining a diamond requires many prerequisite items. Hence it is almost impossible for an agent to obtain a diamond via any random exploration policies.

The MineRL (13) dataset consists of over 60 million state-action-reward tuples of recorded human expert demonstrations over all the seven environments. Each trajectory is contiguously sampled every Minecraft game tick (at 20 game ticks every second). Each state is comprised of an RGB video frame of the player's point-of-view and a comprehensive set of features from the game-state at that tick: player inventory, item collection events, distances to objectives, player attributes (health, level, achievements), and details about the current GUI the player has open. The action recorded at each tick consists of: all the keyboard presses, the change in view pitch and mouse movements, player GUI interactions, and agglomerative actions such as item crafting. Human trajectories are accompanied by automatically generated annotations. All environments include metrics that indicate the quality of the demonstration, such as timestamped rewards, number of no-ops, number of deaths, and total score. Additionally, trajectory meta-data includes timestamped markers for hierarchical labelings; e.g. when a house-like structure is built or certain objectives such as chopping down a tree are met.

## 1.3 Evaluation Metrics and some Baselines

Once following training, the gained agent will be evaluated on the average score over 500 episodes. Scores are computed as the sum of the milestone rewards achieved by the agent in a given episode as outlined in Table 1. A milestone is reached when an agent obtains the first instance of the specified item. Ties are broken by the number of episodes required to achieve the last milestone.

Here are some baseline for **ObtainDiamond**. These results in MineRL-2020 challenge (12) are listed in Table 2. For all these methods, K-means are applied to cluster the action space with the human demonstration data, and then train the agent. The implementations are included into PFRL agents library. According to these baseline and experience, imitation learning is more effective in **ObtainDiamond** environment because human experts can provide efficient samples for the agent.

## 2 Related Work of Imitation Learning

In imitation learning (*IL*) instead of trying to learn from the sparse rewards or manually specifying a reward function, an expert (typically a human) provides us with a set of demonstrations. The agent tries to learn the optimal policy $\pi$ that produce similar behaviors by following, imitating the expert's decisions. The *IL* is composed of the environment, which is essentially a Markov Decision Process (*MDP*). Generally, the environment has a set of states, denoted by $S$, a set of actions $A$, a transition model $P(s'|s, a)$, which presents the probability that an action $a$ in the state $s$ leads to state $s'$) and an unknown $R(s, a)$ reward function. The demonstrations are composed of the state and action sequences, which actually are some demonstration trajectories with the form:

$$\tau_e = \{(s_t, a_t)\} \tag{1}$$

where the actions based on the expert's optimal $\pi$ policy.

Generally, the imitation learning can be roughly split into two main categories behavioral cloning (*BC*) and inverse reinforcement learning (*IRL*).

## 2.1 Behavioral Cloning

The simplest imitation learning algorithms of imitation learning is behaviour cloning (*BC*) (28; 24; 6), which focuses on learning the expert's policy using supervised learning given demonstration trajectories $\tau_e$. The first application of behaviour cloning is ALVINN (21), where a vehicle is equipped with sensors, which learned to map the sensor inputs into steering angles and drive autonomously. More specifically, the demonstrations are divided into state-action pairs and treated as i.i.d. examples and then learned by supervised learning with suitable loss functions. Behavioral cloning has been widely used in the context of autonomous driving (5) and control of aerial vehicles (10). Behavioural cloning can work excellently in some cases where it require only demonstration data to learn an imitation policy directly and have no need for further interaction between the environment and agent. However, some applications break the i.i.d assumption and errors made in different states will be added up. A mistake made by the agent can easily put it into a state that the expert has never visited and the agent has never trained on. In such states, the behaviour is undefined and this can lead to catastrophic failures (23).

## 2.2 Inverse Reinforcement Learning

Another major category of imitation learning method is based on inverse reinforcement learning (*IRL*). The main idea of *IRL* is to learn the reward function of the environment based on expert demonstrations, and then use reinforcement learning to find the optimal policy (25; 19). This can often get better imitation results than direct behavioral cloning.

Instead of directly learning a mapping from states to actions using the demonstration data, *IRL*-based methods iternatively alternate between using the demonstration to infer a hidden reward function and using *RL* with the inferred reward function to learn an imitating policy. Specifically, starting with a group of experts' presentations (assuming these presentations are optimal) and then trying to estimate the parameterized reward function, which would cause the expert's behavior. By repeating the following process until a good enough policy is found: first, updating the reward function parameters and then solving the reinforced learning problem (given the reward function, trying to find the optimal policy). Finally, comparing the newly learned policy with the expert's policy (18).

*IRF*-based techniques have been used for a variety of tasks such as maneuvering a helicopter (1) and object manipulation (8). Using *RL* to optimize the policy given the inferred reward function requires the agent to interact with its environment, which can be costly from a time and safety perspective. Moreover, the*IRL* step typically requires the agent to solve an *MDP* in the inner loop of iterative reward optimization (1; 30), which can be extremely costly from a computational perspective. However, recently, a number of methods have been developed which do not make this requirement (8; 16; 9) . One of these approaches is called generative adversarial imitation from observation (*GAIL*) (16), which uses an architecture similar to generative adversarial networks (GANs) (11), and the associated algorithm can be thought of as trying to induce an imitator state-action occupancy measure that is similar to that of the demonstrator.

The general *IRL* algorithm is the following: Depending on the actual problem, there can be two main methods of *IRL*: the model-based and the model-free methods.

In the model-based case, the reward function is linear. In each iteration, the full RL problem need to be solved, so to be able to do this efficiently, assuming that the environment's (the MDP's) state space is small. Also suppose that the state transition dynamics of the environment is known. This is needed so that our learned policy is compared with the expert's one effectively.

The model-free method is a more general case. Here, supposing that the state space of the MDP is large or continuous, therefore in each iteration, only a single step of the RL problem is solved. In this case, the state transitions dynamics of the environment is unknowable, but assume that a simulator or the environment can be access to. Therefore, comparing our policy to the expert's one is trickier.

In both cases, however, learning the reward function is ambiguous. The reason for this is that many rewards functions can correspond to the same optimal policy (the expert's policy). To this end, the maximum entropy principle proposed by Ziebart (32) can be utilized by selecting the largest entropy trajectory distribution.

## 3  Method

Since we have found that all demonstration samples in **ObtainDiamond** are failed in digging a diamond and the main reward is from **Treechop** and **ObtainIronPickaxe** tasks. Therefore the provided demonstration samples is more about these tasks.

In the environment that can be described with time series, the agent can get an observation $s$ and then take an action $a$ according to a specified policy $\pi(s)$. This can be annotated with the time step index $t$: the agent receives an observation $s_t$, then according to the policy $\pi(s_t)$, the agent chooses action $a_t$. After the action, the agent gets a reward $r_t$ as well as the next $s_{t+1}$. In total, the target for the agent is to find a policy $\pi(s)$ and get the most reward $R = \sum_{t=0}^{T} r_t$ over an episode.

Since we already have some demonstration samples that contains human trajectories, imitating these behaviors will be a good way to get the policy $\pi(s)$. Accordingly, we can train a mapping between received observations from demonstration samples and expert actions. This well-trained mapping will give a certain action according to the observation. Therefore the pipeline will be a classification task: different actions is for different classes while the observation is the input feature that need to be classified.

Now we input an observation $s$ into a deep network model $f(\cdot)$ with an $f(s) \in R^n$ vector. For all $N$ kinds of actions, each of them has a prototype vector $w_a \in R^n$. In (2), the outputs of last fully connection layer of model are applied in softmax function. Here we propose an enhanced mixed Gaussian softmax distribution:

$$\pi(s, a^{(i)}) = p_\pi(a^{(i)}|s) = \frac{e^{\alpha g(s, a^{(i)})}}{\sum_{k=1}^{N} e^{\alpha g(s, a^{(k)})}},  \tag{2}$$

where $g(s, a) = e^{|f(s) - w_a|_2}$ is the RBF kernel that related to the Gaussian distance. $\alpha$ is the scaling parameter. This policy will trained with a modified cross entropy loss function in training:

$$L(s) = \sum_{k=1}^{N} -q(a^{(k)}) \log \pi(s, a^{(k)}).  \tag{3}$$

The $q(a^{(k)})$ is a $\beta$-smoothed target. That means if the expert in demonstration samples chooses $a^{(i)}$ as the action, $q(a^{(i)}) = \beta$. For $k \neq i$, $q(a^{(k)}) = \frac{1-\beta}{N-1}$. Thus there are two parameters $\alpha$ and $\beta$ in our objective function.

## 4  Experiment

More training details and resutls are described in this section. We first introduce action and state space, and the architecture of network with training settings. Then we outperform the proposed EGCE with a high baseline.

### 4.1  Action and State Space

The environment observations are the main components of the states, which actually are $64 \times 64$ RGB images. In the *ObtainIronPickaxe* task, an additional vector state is provided, which contains information about the following: collected resources and hand tools, as well as items currently held. The held is encoded as a one-hot vector, and the items are encoded as multi-hot vectors (the quantity in the sub-vector is equal to the number of corresponding products in the inventory).

For our action space, there are three parts. The first one contains eight binary actions, which is related with the movement in the environment. The eight actions are *forward, backward, left, right, jump,*

*sprint, attack, sneak*. You can use multiple movement actions at the same time step generating 256 combinations. The second one is about the pitch control and continuous yaw of the agent's camera position. The last one is the actions related to the production, equipment and placement of items. Some items needs to be placed on the ground before it can be used such as crafting tables.

Therefore, the simple combination of the actions creates a huge action space. Here we follow (5), who is the first tor propose quantize the continuous control and use 22.5 degrees for each direction.

After quantifying the motion of the camera, there are 1280 possible motion combinations. We only allow up to 3 simultaneous actions, and remove any superfluous actions, such as rotating left and right at the same time. So there are 112 different movements finally. More complete description of motion and state encoding is available in supplementary materials.

### 4.2 Architecture and Training Setup

The proposed strategy neural network consists of three parts. The first part is a convolutional perception for image, the second part is a fully connected layer for the input and the vector part of the state are connected after the last layer. The last layer has softmax or linear output for cross entropy or margin loss. We investigated the relationship between network size and performance by testing three different networks. Architecture of network awareness part: DQN architecture with 3 convolutional layers (17), Impala architecture with 6 remaining blocks (7) and Deep Impala architecture with 8 remaining blocks remaining blocks and Fixup initialization ( (20; 31)). During the last architecture tested, the channel sizes of all convolutional layers and fully connected layers are doubled.

We used Adam optimizer to train the network, with a learning rate of $6.25 \times 10^{-5}$ and a weight attenuation of $10^{-5}$ with maximum $3 \times 106$ steps. From the demo dataset, we used human trajectories to successfully achieve environmental goals within the time limits of the respective tasks (*i.e.*, ObtainDiamond and ObtainIronPickaxe). We also delete all states where humans have not taken any action. For a complete network architecture, please refer to supplementary materials. In addition, we also tested various enhancements, such as contrast, rectangle removal, horizontal flipping (where the left and right actions are also flipped), sharpness, brightness and descendant adjustments. In addition to assessing policy performance, we tested two additional performance indicators: training losses and testing losses on invisible human trajectories to assess their relevance to actual performance policy.

### 4.3 Additional Data Incorporation

We use the human track from the ObtainDiamond and ObtainIronPickaxe tasks in the default training settings. In order to create a more realistic observation of the additional data, we first sample the random states and ObtainDiamond tracks, where the rewards have not yet reached. Then vector observation are used to observe part of the sample state to complete the Treechop observation. Therefore, until whole of the Treechop states are totally observed the process is ended.

### 4.4 Resutls

In this section, we compared imitation learning performance with different losses. Following (2), we record 8 snapshots of the deep impala network (7). Image flipping is the only applied data augmentations. The reward is the average of 40 episodes for each snapshot. Then the best performed snapshot is recorded as the running results. According to the results in Table 3, our proposed enhanced Gaussian cross entropy (EGCE) can obtain the highest reward among these three losses applied in imitation learning: the average reward of EGCE without *TreeChop* data achieves 50.30, which surpass the original cross entropy in (2) about 7.7%. This implies EGCE can actually improve the performance of imitation learning pipeline.

## 5 Conclusion

The MineRL *ObtainDiamon* task is a very challenge domain. Applying imitation learning with demonstration samples can achieve ideal performances. Since our proposed enhanced Gaussian cross-entorpy can significantly improve the reward, this fact implies that weak cor-realtion between

| Methods | Reward | Settings |
|---------|--------|----------|
| Cross Entropy | 46.72 | - |
| Cross Entropy | 65.36 | with *TreeChop* data |
| Margin Loss | 42.48 | - |
| Margin Loss | 34.21 | with *TreeChop* data |
| EGCE (Ours) | 50.30 | $\alpha = 25, \beta = 0.9$ |
| EGCE (Ours) | **66.56** | $\alpha = 25, \beta = 0.9$, with *TreeChop* data |

Table 3: Some results of exist solution and proposed methods.

loss values and policy performance can be partly relieved by a well-designed loss. The proposed EGCE can stabilize the loss and set up a strong connection with policy performance.

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
