# OpenReview forum: "Using Enhanced Gaussian Cross-Entropy in Imitation Learning to Digging the First Diamond in Minecraft"
_CUHK.edu.hk/2021/Course/IERG5350_

### Official Review · AnonReviewer1 · 2020-12-15
**Modified cross-entropy loss to achieve higer reward in ObtainDiamond task.**

**Rating:** 6
**Confidence:** 3

**Review:**

Summary: The paper proposes an enhanced Gaussian cross-entropy loss for imitation learning problems to achieve higher reward in the ObtainDiamond task (based on the MineRL environment). Experiments comparisons show that EGCE achieves a higher reward compared to Cross-entropy baselines.

Overall Comments: The author tackles a complex setting (MineRL) where existing baselines cannot achieve high rewards. They focus on loss design, with a modified cross-entropy loss with two introduced parameters α, β. While the results seem promising, there lacks a strong motivation and supporting claims in their writing paper to the new loss design. I'd be curious to hear more about their explanations on "this fact implies that weak correlation between loss values and policy performance can be partly relieved by a well-designed loss." in their conclusion.

My suggestion is to provide more analysis or explanations on the loss design in the revision version :)

---

### Official Review · AnonReviewer3 · 2020-12-19
**Good Practice in Solving Problem Using Reinforcement Learning with A Notable Enhancement Proposed**

**Rating:** 6
**Confidence:** 3

**Review:**

In this report, the authors propose to integrate the Gaussian mixture with the final softmax function of the policy model, aiming at improving the outcome of imitation learning. The challenging final task along with its prerequisites are well described in this report. The problem sounds so difficult to solve that I would barely touch it by spending a few more months in studying reinforcement. learning. It's impressive that the authors solve the task in this course project and achieve a better result compared with the baseline.

Generally, the report provides enough information to illustrate the authors' ideal on how to improve the baseline.  The result also provides convincing evidence that the authors surely put in their efforts to accomplish the project. However, the way they describing the experiment seems to be slightly rough. More details about their implementation are expected to evaluate their work. Several concerns might help the authors revise this report:

(1)  In the experiment, it seems that the authors don't explicitly state the imitation algorithm used to solve the task. Is there any paper I can refer to?

(2) In section 4.2, there mentions a supplementary material, which is missing from this version. The missing supplementary material does hurt the completeness of this report.

(3) Figure 1 is taken from the NeurlIPS MineRL Competition 2020 webpage. It's recommended to put on the citation or make a few changes to the figure. Mabe, add a sequence number can better illustrate the order of those subtasks.

(4) The result would become more solid if certain graphs or plots are provided, such as the varying loss during the training.

(5) In the introduction paragraph of section 4, what's the meaning of "Then we outperform the proposed EGCE with a high baseline."?

Overall, the courage of the authors tacking this competition is quite impressive. If they participated in the competition, I hope the authors have got a good ranking in the competition. Although, this report needs some amendment to exhibit their efforts in a more concise way.

---

### Official Review · AnonReviewer2 · 2020-12-20
**Summary: This paper focuses on improving the performance of imitation learning problem. The main contribution of this paper is to use an enhanced Gaussian cross entropy loss for imitation learning problem based on the MineRL environment.**

**Rating:** 7
**Confidence:** 3

**Review:**

Originality: This paper modifies cross entropy loss function and turns the imitation learning problem into a classification problem.
Quality: The paper's technical quality is good. Rewards are clearly defined and the environment settings are mentioned.
Clarity: This paper is clearly written and the sections are organized in a logical way. Figures and tables in the paper are clear and informative.
Suggestion: I suggest the authors to provide more comparison experiments on how to choose the two parameters of loss function.